# ON THE EQUIVALENCE OF GRAPH CONVOLUTION AND MIXUP

## ABSTRACT

This paper investigates the relationship between graph convolution and Mixup techniques. Graph convolution in a graph neural network involves aggregating features from neighboring samples to learn representative features for a specific node or sample. On the other hand, Mixup is a data augmentation technique that generates new examples by averaging features and one-hot labels from multiple samples. One commonality between these techniques is their utilization of information from multiple samples to derive feature representation. This study aims to explore whether a connection exists between these two approaches. Our investigation reveals that, under two mild conditions, graph convolution can be viewed as a specialized form of Mixup that is applied during both the training and testing phases. The two conditions are: 1) *Homophily Relabel* - assigning the target node's label to all its neighbors, and 2) *Test-Time Mixup* - Mixup the feature during the test time. We establish this equivalence mathematically by demonstrating that graph convolution networks (GCN) and simplified graph convolution (SGC) can be expressed as a form of Mixup. We also empirically verify the equivalence by training an MLP using the two conditions to achieve comparable performance. The code can be found at https://anonymous.4open.science/r/GraphConv_is_Mixup-F470.

## 1 INTRODUCTION

Graph Neural Networks (GNNs) (Wu et al., 2020; Zhou et al., 2020) have recently been recognized as the *de facto* state-of-the-art algorithm for graph learning. The core idea behind GNNs is neighbor aggregation, which involves combining the features of a node's neighbors. Specifically, for a target node with feature $\mathbf{x}_i$, one-hot label $\mathbf{y}_i$, and neighbor set $\mathcal{N}_i$, the graph convolution operation in GCN is essentially as follows:

$$(\tilde{\mathbf{x}}, \tilde{\mathbf{y}}) = \left( \frac{1}{|\mathcal{N}_i|} \sum_{k \in \mathcal{N}_i} \mathbf{x}_k, \ \mathbf{y}_i \right), \tag{1}$$

In parallel, Mixup (Zhang et al., 2018) is proposed to train deep neural networks effectively, which also essentially generates a new sample by averaging the features and labels of multiple samples:

$$(\tilde{\mathbf{x}}, \tilde{\mathbf{y}}) = \left( \sum_{i=1}^{N} \lambda_i \mathbf{x}_i, \ \sum_{i=1}^{N} \lambda_i \mathbf{y}_i \right), \quad \text{s.t.} \quad \sum_{i=1}^{N} \lambda_i = 1, \tag{2}$$

where $\mathbf{x}_i / \mathbf{y}_i$ are the feature/label of sample $i$. Mixup typically takes two data samples ($N = 2$).

Equation (1) and Equation (2) highlight a remarkable similarity between graph convolution and Mixup, i.e., *the manipulation of data samples through averaging the features*. This similarity prompts us to investigate the relationship between these two techniques as follows:

*Is there a connection between graph convolution and Mixup?*

In this paper, we answer this question by establishing the connection between graph convolutions and Mixup, and further understanding the graph neural networks through the lens of Mixup. We show that graph convolutions are intrinsically equivalent to Mixup by rewriting Equation (1) as follows:

$$(\tilde{\mathbf{x}}, \tilde{\mathbf{y}}) = \left( \frac{1}{|\mathcal{N}_i|} \sum_{k \in \mathcal{N}_i} \mathbf{x}_k, \mathbf{y}_i \right) = \left( \sum_{k \in \mathcal{N}_i} \frac{1}{|\mathcal{N}_i|} \mathbf{x}_k, \sum_{k \in \mathcal{N}_i} \frac{1}{|\mathcal{N}_i|} \mathbf{y}_i \right) \stackrel{\lambda_i = \frac{1}{|\mathcal{N}_i|}}{=} \left( \sum_{k \in \mathcal{N}_i} \lambda_i \mathbf{x}_k, \sum_{k \in \mathcal{N}_i} \lambda_i \mathbf{y}_i \right),$$

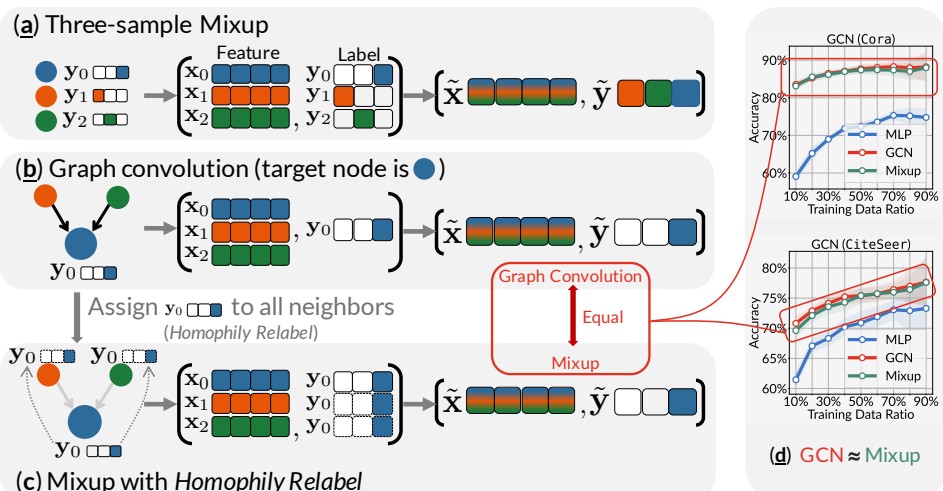

Figure 1: Graph convolution is Mixup. (**a**) illustrates the basic idea of Mixup: averaging the features and one-hot labels of multiple samples (●, ●, ●). (**b**) shows the graph convolution operation where the feature of the target node (●) is the weighted average of the features of all its neighbors. (**b**) → (**c**) shows that graph convolution is Mixup if we assign the label (▢▢◼) of the target node (●) to all of its neighbors (●, ●). (**d**) shows that Mixup is empirically equivalent to GCN.

where $\mathbf{x}_i$ and $\mathbf{y}_i$ are the feature and label of the target node $n_i$. This equation states that graph convolution is equivalent to Mixup if we assign the $\mathbf{y}_i$ to all the neighbors of node $n_i$ in set $\mathcal{N}_i$

To demonstrate the equivalence between graph convolutions and Mixup, we begin by illustrating that a one-layer graph convolutional network (GCN) (Kipf & Welling, 2016b) can be transformed into an input Mixup. A two-layer GCN can be expressed as a hybrid of input and manifold Mixup (Verma et al., 2019). Similarly, simplifing graph convolution (SGC) (Wu et al., 2019) can be reformulated as an input Mixup. We thus establish the mathematical equivalence between graph convolutions and Mixup, under two mild and reasonable conditions: 1) assign the target node's label to neighbors in the training time (referred to as *Homophily Relabel*); 2) perform feature mixup in the test time (referred to as *Test-Time Mixup*).

We further investigate the conditions required for the equivalence between graph convolution and Mixup, focusing on the effect of *Homophily Relabel* and *Test-Time Mixup*. To explore *Homophily Relabel*, we demonstrate that training an MLP with *Homophily Relabel* (called HMLP) is equivalent to training GCN in terms of prediction accuracy. This finding provides a novel perspective for understanding the relationship between graph convolution and Mixup and suggests a new direction for designing efficient GNNs. To investigate Test-Time Mixup, we train GNNs without connection information, perform neighbor aggregation during inference, and find that this approach can achieve performance comparable to traditional GNNs. This result reveals that Test-Time Mixup can be a powerful alternative to traditional GNNs in some scenarios, and suggests that Mixup may have a broader range of applications beyond traditional data augmentation. Our investigation on *Homophily Relabeling* and *Test-Time Mixup* provides valuable insights into the theoretical properties of GNNs and their potential applications in practice. We highlight our **contributions** as follows:

- We establish for the first time connection between graph convolution and Mixup, showing that graph convolutions are mathematically and empirically equivalent to Mixup. This simple yet novel finding potentially opens the door toward a deeper understanding of GNNs.

- Given that both graph convolution and Mixup perform a weighted average on the features of samples, we reveal that graph convolution is conditionally equivalent to Mixup by assigning the label of a target node to all its neighbors during training time (*Homophily Relabel*). Also, we also reveal that in the test time, graph convolution is also Mixup (*Test-Time Mixup*).

- Based on *Homophily Relabel* and *Test-Time Mixup*, we propose two variants of MLPs based on the Mixup strategy, namely HMLP and TMLP, that can match the performance of GNNs.

**Related Work.** Graph neural networks are widely adopted in various graph applications, including social network analysis (Fan et al., 2019), recommendation (Wei et al., 2022; Deng et al., 2022; Cai

et al., 2023; Tang et al., 2022), knowledge graph (Cao et al., 2023; Zhao et al., 2023), molecular analysis (Sun et al., 2022; Zhu et al., 2023a; ZHANG et al., 2023; Wang et al., 2023b; Corso et al., 2023; Liao & Smidt, 2023; Xia et al., 2023; Hladiš et al., 2023), drug discovery (Sun et al., 2022; Zhang et al., 2023c), link prediction (Chamberlain et al., 2023; Cong et al., 2023) and others (Chen et al., 2023a). Understanding the generalization and working mechanism of graph neural networks is still in its infancy (Garg et al., 2020; Zhang et al., 2023a; Yang et al., 2023; Baranwal et al., 2023). The previous work attempts to understand graph neural networks from different perspectives, such as signal processing (Nt & Maehara, 2019; Bo et al., 2021; Bianchi et al., 2021), gradient flow (Di Giovanni et al., 2022), dynamic programming(Dudzik & Veličković, 2022), neural tangent kernels (Yang et al., 2023; Du et al., 2019; Sabanayagam et al., 2022) and influence function (Chen et al., 2023b). There is also a line of works that analyzes the connection between GNNs and MLPs (Baranwal et al., 2023; Han et al., 2022b; Yang et al., 2023; Tian et al., 2023), which is similar to the proposed method in Section 3. In this work, we understand graph neural networks through a fresh perspective, Mixup. We believe that this work will inspire further research and lead to the development of new techniques to improve the performance and interpretability of GNNs.

In parallel, Mixup (Zhang et al., 2018) and its variants (Verma et al., 2019; Yun et al., 2019; Kim et al., 2020; 2021) have emerged as a popular data augmentation technique that improves the generalization performance of deep neural networks. Mixup is used to understand or improve many machine learning techniques. More specifically, deep neural networks trained with Mixup achieve better generalization (Chun et al., 2020; Zhang et al., 2020; Chidambaram et al., 2022), calibration (Thulasidasan et al., 2019; Zhang et al., 2022a), and adversarial robustness (Pang et al., 2019; Archambault et al., 2019; Zhang et al., 2020; Lamb et al., 2019). Mixup are also used to explain ensmeble (Lopez-Paz et al., 2023). See more related work in Appendix A.

**The Scope of Our Work.** This work not only provides a fresh perspective for comprehending Graph Convolution through Mixup, but also makes valuable contributions to the practical and theoretical aspects of graph neural networks, facilitating efficient training and inference for GNNs when dealing with large-scale graph data. It's important to note that certain complex GNN architectures, such as attention-based graph convolution (Veličković et al., 2018; Xu et al., 2018; Javaloy et al., 2023), may not lend themselves to a straightforward transformation into a Mixup formulation due to the intricacies of their convolution mechanisms. In our discussion, we explore how these GNNs can also be considered as a *generlized* Mixup in Section 2.4.

## 2   GRAPH CONVOLUTION IS MIXUP

In this section, we reveal that the graph convolution is essentially equivalent to Mixup. We first present the notation used in this paper. Then we present the original graph convolution network (GCN) (Kingma & Ba, 2015) and simplifying graph convolutional networks (SGC) (Wu et al., 2019) can be expressed mathematically as a form of Mixup. Last, we present the main claim, i.e., graph convolution can be viewed as a special form of Mixup under mild and reasonable conditions.

**Notations.** We denote a graph as $\mathcal{G}(\mathcal{V}, \mathcal{E})$, where $\mathcal{V}$ is the node set and $\mathcal{E}$ is the edge set. The number of nodes is $N = |\mathcal{V}|$ and the number of edges is $M = |\mathcal{E}|$. We denote the node feature matrix as $\mathbf{X} = \{\mathbf{x}_1, \mathbf{x}_2, \cdots, \mathbf{x}_N\} \in \mathbb{R}^{N \times d}$, where $\mathbf{x}_i$ is the node feature for node $n_i$ and $d$ is the dimension of features. We denote the binary adjacency matrix as $\mathbf{A} \in \{0,1\}^{N \times N}$, where $\mathbf{A}_{ij} = 1$ if there is an edge between node $n_i$ and node $n_j$ in edge set $\mathcal{E}$, and $\mathbf{A}_{ij} = 0$ otherwise. We denote the neighbor set of node $n_i$ as $\mathcal{N}_i$, and its 2-hop neighbor set as $\mathcal{N}_i^2$. For the node classification task, we denote the prediction targets of nodes as one-hot labels $\mathbf{Y} \in \{0,1\}^{N \times (C-1)}$, where $C$ is the number of classes. For graph convolution network, we use $\tilde{\mathbf{A}} = \mathbf{D}^{-1}\mathbf{A}$ as the normalized adjacency matrix[1], where $\mathbf{D} \in \mathbb{R}^{N \times N}$ is the diagonal degree matrix of $\mathbf{A}$, and $\mathbf{D}_{ii}$ denotes the degree of node $n_i$. We use $\tilde{\mathbf{a}}_i \in \mathbb{R}^{1 \times N}$, $i$-th row of $\tilde{\mathbf{A}}$, as the normalized adjacency vector of node $n_i$.

### 2.1   PRELIMINARIES

**Graph Convolution.** Graph Convolution Network (GCN) (Kipf & Welling, 2016a), as the pioneering work of GNNs, proposes $\hat{\mathbf{Y}} = \text{softmax}(\tilde{\mathbf{A}} \cdot \sigma(\tilde{\mathbf{A}} \cdot \mathbf{X} \cdot \mathbf{W}_1) \cdot \mathbf{W}_2)$, where $\mathbf{W}_1$ and $\mathbf{W}_2$ are the

---

[1]Our following analysis can be easily generalized to other normalization methods, such as $\mathbf{D}^{-\frac{1}{2}}\mathbf{A}\mathbf{D}^{-\frac{1}{2}}$.

trainable weights of the layer one and two, respectively. Simplifying Graph Convolutional Networks (SGC) (Wu et al., 2019) is proposed as $\hat{\mathbf{Y}} = \text{softmax}(\tilde{\mathbf{A}} \cdot \tilde{\mathbf{A}} \cdot \mathbf{X} \cdot \mathbf{W})$. In this work, we take these two widely used GNNs to show that graph convolution are essentially Mixup.

**Mixup.** The Mixup technique, introduced by Zhang et al. (2018), is a simple yet effective data augmentation method to improve the generalization of deep learning models. The basic idea behind Mixup is to blend the features and one-hot labels of a random pair of samples to generate synthetic samples. The mathematical expression of the two-sample Mixup is as follows:

$$(\tilde{\mathbf{x}}, \tilde{\mathbf{y}}) = (\lambda \mathbf{x}_i + (1 - \lambda)\mathbf{x}_j, \lambda \mathbf{y}_i + (1 - \lambda)\mathbf{y}_j) = (\lambda_1 \mathbf{x}_i + \lambda_2 \mathbf{x}_j, \lambda_1 \mathbf{y}_i + \lambda_2 \mathbf{y}_j), \quad (3)$$

where $\lambda_i + \lambda_j = 1$. Based on the two-sample Mixup, the multiple-sample Mixup is presented in Equation (2). The mathematical expression presented above demonstrates that Mixup computes a weighted average of the features from multiple original samples to generate synthetic samples.

## 2.2 CONNECTING GRAPH CONVOLUTION AND MIXUP

We demonstrate that graph convolution, using GCN and SGC as examples, is conditionally Mixup. To do this, we mathematically reformulate the expressions of GCN and SGC to a Mixup form, thereby illustrating that graph convolutions are indeed Mixup.

**One-layer GCN is Mixup** We begin our analysis by examining a simple graph convolution neural network (Kipf & Welling, 2016a) with one layer, referred to as the one-layer GCN. We demonstrate that the one-layer GCN can be mathematically understood as an implementation of the input Mixup technique. The expression of the one-layer GCN is given by: $\hat{\mathbf{Y}} = \text{softmax}(\tilde{\mathbf{A}} \cdot \mathbf{X} \cdot \mathbf{W})$ where $\mathbf{W} \in \mathbb{R}^{D \times C}$ is the trainable weight matrix. Let us focus on a single node $n_i$ in the graph. The predicted one-hot label for this node is given by $\hat{\mathbf{y}}_i = \text{softmax}(\tilde{\mathbf{a}}_i \cdot \mathbf{X} \cdot \mathbf{W})$, where $\tilde{\mathbf{a}}_i \in \mathbb{R}^{1 \times N}$, $i$-th row of $\tilde{\mathbf{A}}$, is the normalized adjacency vector of node $n_i$. $\mathcal{N}_i$ is the neighbors set of node $n_i$. We make the following observations:

- $\tilde{\mathbf{a}}_i \cdot \mathbf{X}$ results in a weighted sum of the features of the neighbors of node $n_i$, which is the multiple-sample Mixup. Explicitly, we can rewrite $\tilde{\mathbf{a}}_i \cdot \mathbf{X}$ to $\tilde{\mathbf{x}} = \frac{1}{|\mathcal{N}i|} \sum_{k \in \mathcal{N}_i} \mathbf{x}_k$.

- If we assign node $n_i$'s label to all its neighbors, the label $\mathbf{y}_i$ of node $n_i$ can be interpreted as a weighted average of the labels of its neighboring nodes, which is equivalent to performing Mixup on the labels. Thus we have $\tilde{\mathbf{y}} = \mathbf{y}_i = \frac{1}{|\mathcal{N}_i|} \sum_{k \in \mathcal{N}_i} \mathbf{y}_k$.

From the previous observations, we can see that one-layer GCN synthesizes a new Mixup sample for node $n_i$ by mixing up its neighbors' features and one-hot labels, which is defined as follows:

$$(\tilde{\mathbf{x}}, \tilde{\mathbf{y}}) = \left( \frac{1}{|\mathcal{N}_i|} \sum_{k \in \mathcal{N}_i} \mathbf{x}_k, \frac{1}{|\mathcal{N}_i|} \sum_{k \in \mathcal{N}_i} \mathbf{y}_k \right). \quad (4)$$

Therefore, we conclude that *a one-layer GCN is a Mixup machine, which essentially synthesizes a new sample by averaging its neighbors' features and one-hot labels.*

**Two-layer GCN is Mixup** We extend our analysis to consider a two-layer GCN. The expression for two-layer GCN is given by $\hat{\mathbf{Y}} = \text{softmax}(\tilde{\mathbf{A}} \cdot \sigma(\tilde{\mathbf{A}} \cdot \mathbf{X} \cdot \mathbf{W}_1) \cdot \mathbf{W}_2)$, Let us focus on node $n_i$ in the graph. For the first layer of the two-layer GCN, we have $\mathbf{h}_i = \text{Relu}(\tilde{\mathbf{a}}_i \cdot \mathbf{X} \cdot \mathbf{W}_1) \in \mathbb{R}^{1 \times d}$, where $\tilde{\mathbf{a}}_i \in \mathbb{R}^{1 \times N}$ is the normalized adjacency vector of node $n_i$. The first layer is the same as the one-layer GCN as discussed above, which can be regarded as a multiple-sample input Mixup.

For the second layer of the two-layer GCN, we have $\hat{\mathbf{y}}_i = \text{Softmax}(\tilde{\mathbf{a}}_i \mathbf{H} \mathbf{W}_2) \in \mathbb{R}^{1 \times C}$, $\mathbf{H} \in \mathbb{R}^{N \times d}$ is the hidden representation of all nodes obtained from the first layer, and $\mathbf{W}_2 \in \mathbb{R}^{d \times C}$ is the weight matrix. The predicted one-hot label for node $n_i$, $\hat{\mathbf{y}}_i$, is obtained through a softmax activation function. The 2-nd layer can be regarded as multiple-sample manifold Mixup (Verma et al., 2019) in the following:

- $\tilde{\mathbf{a}}_i \cdot \mathbf{H}$ is the multiple-sample Mixup of the hidden representation of the neighbors of node $n_i$. We rewrite the $\tilde{\mathbf{a}}_i \cdot \mathbf{H}$ to $\tilde{\mathbf{x}} = \tilde{\mathbf{a}}_i \cdot \mathbf{H} = \frac{1}{|\mathcal{N}_i|} \sum_{k \in \mathcal{N}_i} \mathbf{h}_k$.

Therefore, we conclude that *a multi-layer GCN is a hybrid of input Mixup (first layer) and manifold Mixup (second layer).*

**SGC is Mixup** The 2-layer SGC architecture is represented by the following expression, $\hat{\mathbf{Y}} = \text{softmax}(\tilde{\mathbf{A}} \cdot \tilde{\mathbf{A}} \cdot \mathbf{X} \cdot \mathbf{W})$. Similar to one-layer GCN, for the node $n_i$ in a 2-layer SGC, we have

$$\hat{\mathbf{y}}_i = \text{softmax}(\tilde{\mathbf{a}}_i^2 \cdot \mathbf{X} \cdot \mathbf{W}), \mathcal{L} = \text{Cross-Entropy}(\hat{\mathbf{y}}_i, \mathbf{y}_i), \tag{5}$$

where $\tilde{\mathbf{a}}_i^2 \in \mathbb{R}^{1 \times N}$ is the adjacency vector within 2-hop neighbors of node $n_i$. The 2-hop neighbor set of node $n_i$ is represented by $\mathcal{N}_i^2$, Hereby we have

- $\tilde{\mathbf{a}}_i^2 \cdot \mathbf{X}$ is the multiple Mixup of the features of the neighbors of node $n_i$. We rewrite the $\tilde{\mathbf{a}}i \cdot \mathbf{X}$ to $\tilde{\mathbf{x}} = \tilde{\mathbf{a}}_i^2 \cdot \mathbf{X} = \frac{1}{|\mathcal{N}_i^2|} \sum k \in \mathcal{N}_i^2 \mathbf{x}_k$.

From the above, we have a Mixup of samples for node $n_i$ is

$$(\tilde{\mathbf{x}}, \tilde{\mathbf{y}}) = \left( \frac{1}{|\mathcal{N}_i^2|} \sum_{k \in \mathcal{N}_i^2} \mathbf{x}_k, \quad \frac{1}{|\mathcal{N}_i^2|} \sum_{k \in \mathcal{N}_i^2} \mathbf{y}_k \right). \tag{6}$$

Thus we conclude that *an SGC is an input Mixup machine*.

## 2.3 GRAPH CONVOLUTION IS (*Conditionally*) MIXUP

It is straightforward to derive the Mixup form of 1-layer, 2-layer GCN, and SGC as discussed above. This leads to the conclusion that 1-layer, 2-layer GCN, and SGC can all be reformulated in the form of Mixup. This establishes a mathematical connection between graph neural networks and the Mixup method. Building on these findings, in this section, we introduce our main contribution as follows:

> **Main Results**
>
> Graph convolution is Train- and Test-Time Mixup, under two mild and reasonable conditions: *Homophily Relabel* and *Test-time Mixup*.

The two primary differences between GNNs and Mixup are as follows:

- **Homophily Relabel** In the training time, if we assign the label of the target node to all its neighbors, the Graph Neural Network can be naturally rewritten as a form of Mixup.

- **Test-Time Mixup** In the test time, the GNNs perform feature mixing on the nodes and then use the mixed feature for the inference.

Both of the above conditions are mild and reasonable and have practical implications for GNN design and analysis. The *Homophily Relabel* operation can be understood as imposing the same label on the target node and all its neighbors, which corresponds to the homophily assumption for graph neural networks. The homophily assumption posits that nodes with similar features should have the same label, which is a common assumption in many real-world graph applications. On the other hand, *Test-time Mixup* can be understood as Mixup at the test time by mixing the features from neighbors for prediction. We examine the above differences in depth in Section 3 and Section 4.

## 2.4 DISCUSSION

**Comparison to Other Work.** Hereby, we compare our work and previous work. Yang et al. (2023); Han et al. (2022b) propose to train an MLP and transfer the converged weight from MLP to GNN, which can achieve comparable performance to GCN. In this work, these two methods are all included in part of our work, *Test-Time Mixup*. Different from previous work, we provide a fresh understanding of this phoneme by connecting graph convolution to Mixup and also derive a TMLP Appendix C.2 to implement *Test-Time Mixup*. Baranwal et al. (2022) understand the effect of graph convolution by providing a rigorous theoretical analysis based on a synthetic graph. Different form this work, our work understands graph convolution with a well-studied technique, Mixup.

## 3 IS *Homophily Relabel* EQUIVALENT TO GCNs TRAINING ?

In this section, we conduct experiments to examine the *Homophily Relabel* proposed in Section 2.3. The empirical evidence substantiates our claims, emphasizing the significant effect of Homophily

Relabel on GNNs. This understanding can facilitate the design of an MLP equivalent that competes with the performance of GNNs. Note that we explore the transductive setting in our experiments.

## 3.1 *Homophily Relabel* YIELDS A TRAINING-EQUIVALENT MLP TO GCN

We utilize the simple example graph to demonstrate the calculation of the loss function in a GCN in the transductive setting. The example graph, in Figure 2, comprises three nodes. The blue and red nodes belong to the training set, while the gray node belongs to the test set. In the transductive scenario, the loss function is computed for the entire graph during the training phase, incorporating both the training and test nodes. In our simple example graph, the loss function will be calculated on the blue and red nodes with their actual labels. Notably, the prediction for the gray node, despite it being part of the test set, will also contribute to the overall loss calculation in the transductive setting. For node $n_i$, the prediction $(p_0, p_1)$, the loss of the example graph will be as follows

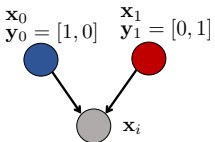

Figure 2: The example graphs. $\mathbf{x}_i$ is the target node, The loss of $\mathbf{x}_i$ that connected two nodes with different labels.

$$
\begin{aligned}
L_{\mathcal{G}} &= \text{cross-entropy}(\mathbf{Y}, \hat{\mathbf{Y}}) = \text{cross-entropy}(\mathbf{y}_0, \hat{\mathbf{y}}_0) + \text{cross-entropy}(\mathbf{y}_1, \hat{\mathbf{y}}_1) \\
&= -y_{00} \log(p_0) - y_{01} \log(p_1) - y_{10} \log(p_0) - y_{11} \log(p_1) \\
&= -1 * \log(p_0) - 0 * \log(p_1) - 0 * \log(p_0) - 1 * \log(p_1) \\
&= -1 * \log(p_0) - 1 * \log(p_1) = -2 * (0.5 * \log(p_0) + 0.5 * \log(p_1)).
\end{aligned}
\tag{7}
$$

where $y_{00}/y_{01}$ is the 0th/1st elements in the one-hot label $\mathbf{y}_0$ for node $n_0$. The above analysis shows that the actual training label for target node $n_i$ is $(0.5, 0.5)$. From the Mixup perspective, in an ideally well-trained model utilizing a two-sample Mixup, the following approximation will hold $f(\lambda_k x + (1 - \lambda_k)x_k) \approx \lambda_k f(x) + (1 - \lambda_k)f(x_k)$, If we regard the neighbor aggregation as Mixup, we can easily derive that

$$
\hat{\mathbf{y}}_0 = f\left(\frac{\mathbf{x}_0 + \mathbf{x}_i}{2}\right) = \frac{1}{2}\left(f(\mathbf{x}_0) + f(\mathbf{x}_i)\right), \hat{\mathbf{y}}_1 = f\left(\frac{\mathbf{x}_1 + \mathbf{x}_i}{2}\right) = \frac{1}{2}\left(f(\mathbf{x}_1) + f(\mathbf{x}_i)\right). \tag{8}
$$

From the above two equations, we can see that $f(x_i)$ are trained with the labels of all its neighbors $\mathbf{x}_0$ and $\mathbf{x}_1$. Thus in the next, *we propose explicitly training the nodes in the test set with the label of their neighbors if they are in the training set.* In our example graph, we explicitly train the gray node with label $(0.5, 0.5)$

**HMLP**  Based on the above analysis, we proposed Homophily Relabel MLP (HMLP), which achieves comparable performance of GCN for via training a MLP as a backbone with *Homophily Relabel*. In detail, the proposed method HMLP has two major steps: 1) relabel all the nodes in the graph with the mixed label $\hat{\mathbf{Y}} = \mathbf{A} \cdot \mathbf{Y}$. 2) train an MLP on all the nodes with $\hat{\mathbf{Y}}$ on relabeled graph, only using node features. We illustrate our proposed HMLP with the example graph in Figure 2. 1) we relabel all the nodes using *Homophily Relabel*, the node with new label will be $(\mathbf{x}_0, (1, 0)), (\mathbf{x}_1, (0, 1)), (\mathbf{x}_i, (0.5, 0.5))$ 2) then train the HMLP with these data samples. The test sample will be $(\mathbf{x}_i, ?)$ during the test time. Note that the backbone of HMLP is an MLP.

## 3.2 CAN HMLP ACHIEVE COMPARABLE PERFORMANCE TO GCN?

To verify the proposed HMLP can achieve comparable performance to the original GCN, we train the one-layer GCN (GCN-1), two-layer GCN (GCN-2), SGC, and HMLP on the training set and report the test accuracy based on the highest accuracy achieved on the validation set. We experimented with different data splits of train/validation/test (the training data ratio span from $10\% - 90\%$), and we also conducted experiments with the public split on Cora, CiteSeer, and PubMed datasets. We present the results in Table 1 and Figure 4. Additionally, we also present the training and test curves of the MLP. GCN and HMLP in Figure 3.

Table 1: Performance comparison of GCN and HMLP.

| Datasets | MLP | GCN | HMLP |
|---|---|---|---|
| Cora | $73.57_{\pm0.98}$ | $88.04_{\pm1.26}$ | $86.42_{\pm1.78}$ |
| CiteSeer | $71.90_{\pm1.69}$ | $75.69_{\pm1.36}$ | $75.94_{\pm1.54}$ |
| PubMed | $86.90_{\pm0.74}$ | $87.94_{\pm0.64}$ | $88.34_{\pm0.48}$ |
| Average | 77.45 | **83.89** | **83.57** |

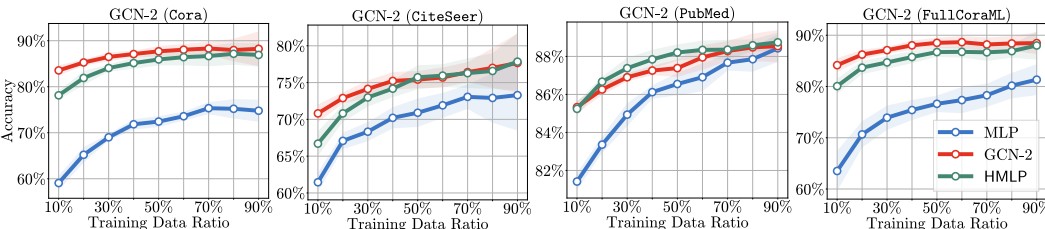

Figure 4: The performance comparison of the GCN, MLP and HMLP (Ours). The x-axis represents the ratio of training data, and the y-axis represents the accuracy of classification. The results show that our proposed method (HMLP) achieves comparable performance to GCN. Note that the architecture of our method in train and test time are both MLP. More experimental results on other datasets and GNNs are presented in Appendix C.1.

From the results in Table 1 and Figure 4, ❶ **HMLP achieves comparable performance to GCN when the data training data split ratio is large.** For the Figure 4, the performance of HMLP is significantly better than MLP, and on par with GCN, especially when the training data ratio is large. The results also show that when the training data ratio is small, HMLP performs worse than GCN. The training and test curves show that HMLP achieve the same test curves (red and green lines in the right subfigure) even though the training curve are not the same. This may be because the labels of nodes are changed.

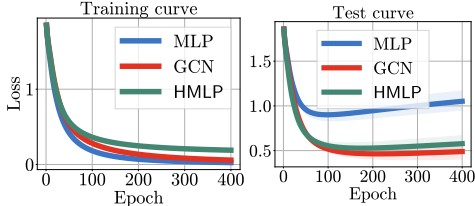

Figure 3: The training and test curves of GCN and HMLP.

## 4 How Does *Test-time Mixup* Affect GCN Inference?

In this section, we aim to investigate how *Test-time Mixup* affects the GCN inference process. We conduct an empirical study to understand the functionality of *Test-time Mixup* (neighbor aggregation during the test time). To this end, we propose an MLP involving *Test-time Mixup* only (TMLP).

**TMLP** The proposed method TMLP follows the steps: 1) we train an MLP using only the node features of the training set. 2) we employ the well-trained MLP for inference with neighbor aggregation during testing. This paradigm has also been investigated in previous works (Han et al., 2022b; Yang et al., 2023). We call this method TMLP. We illustrate the TMLP with the example graph in Figure 2. Specifically, we train the following training examples, $(\mathbf{x}_0, (1, 0)), (\mathbf{x}_1, (0, 1))$. Note that the backbone of TMLP is an MLP. In the following, we explore the performance of TMLP.

### 4.1 How TMLP Perform Compared to GCN?

In the section, we conducted experiments to verify the effect of *Test-Time Mixup*. In the experiments, during the training, we only use the node feature to train an MLP, and then during the test time, we use the well-trained MLP with the neighbor aggregation to perform the inference. We present the results of varying training data ratio in Figure 6, and we also present the accuracy in Table 2 when the training data ratio is 0.6. Additionally, we compare the learned representation of GNNs and TMLP using t-SNE Van der Maaten & Hinton (2008). From these results, we make the following observations:

From these results, we make the following observations: ❷ **TMLP almost achieves comparable performance to GNNs.** In the Figure 4, the performance of TMLP is significantly better than MLP and on par with GCN, especially when the training data ratio is large. In some datasets (i.e., Cora, FullCoraML), the performance of TMLP is slightly worse than GCN but still significantly better than MLP. It is worth noting that our proposed TMLP is only trained with the node

Table 2: Performance comparison of GCN and HMLP.

| Datasets | MLP | GCN | TMLP |
|---|---|---|---|
| Cora | $73.57_{\pm 0.98}$ | $88.04_{\pm 1.26}$ | $88.26_{\pm 1.60}$ |
| CiteSeer | $71.90_{\pm 1.69}$ | $75.69_{\pm 1.36}$ | $76.35_{\pm 1.13}$ |
| PubMed | $86.90_{\pm 0.74}$ | $87.94_{\pm 0.64}$ | $87.58_{\pm 0.44}$ |
| Average | $77.45$ | **83.89** | **84.06** |

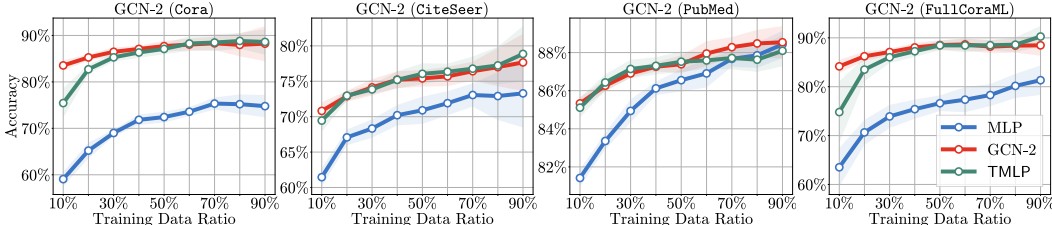

Figure 6: The performance comparison of the GCNs, MLP and TMLP (Ours). GCN-1 is a one-layer GCN, and GCN-2 is a two-layer GCN. The results show that TMLP achieves a comparable performance to GCN and SGC. Note that the architecture is essentially MLP in training time and GCN in the test time. More results on other datasets and GNNs are presented in Appendix C.2.

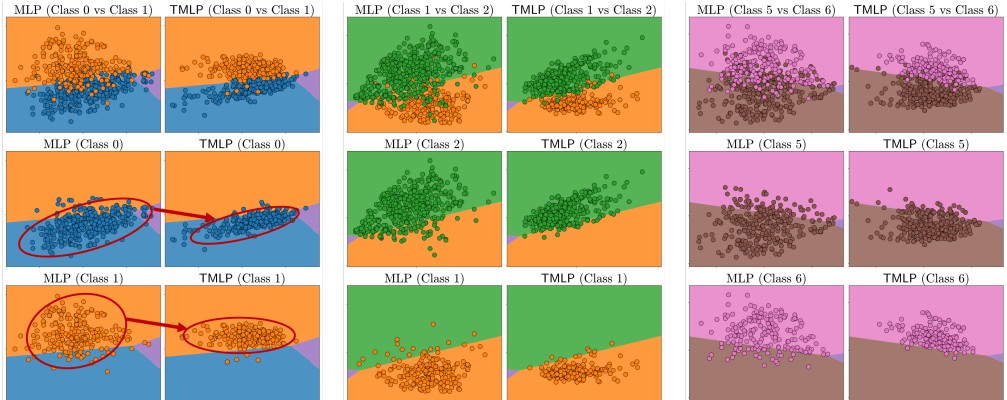

Figure 7: Decision Boundary of MLP and TMLP. TMLP can cause node features to move away from the learned boundary, as shown with ◯→◯. More experimental details are in Appendix C.3

features, ignoring the connection information. Table 2 show the average accuracy (84.06) of TMLP is comparable to that of GCN (83.89).

Based on the node representation, ❸ *Test-Time Mixup* **make the node representation more discriminative than MLP.** The visualization clearly illustrates that the node representations within the same class, as learned by TMLP, are clustered together. This clustering pattern demonstrates the advanced predictive power of TMLP, as it appears to have a keen ability to distinguish similar group classes effectively. Consequently, this capacity allows TMLP to deliver superior classification performance. The node representation clearly show that what TMLP does from the representation perspective.

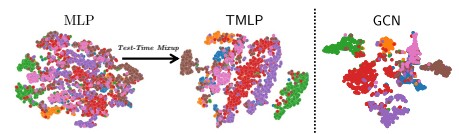

Figure 5: Visualization of node representations learned by MLP and TMLP. Node representations from the same class, after *Test-Time Mixup*, become clustered together.

### 4.2 HOW DOES *Test-time Mixup* BENEFIT GCN? INVESTIGATING DECISION BOUNDARY

In this section, we evaluate the effect of *Test-Time Mixup* on the decision boundary change from MLP to TMLP. To do so, we compare the distance between node features and the decision boundary of MLP before and after applying *Test-Time Mixup*. After training a model with training-time data mixup, we evaluate its decision boundary, which separates different classes in the feature space. We then apply test-time mixup techniques to node features and reassess the decision boundary. Our findings indicate that ❹ *Test-Time Mixup* **can cause node features to move away from the learned boundary**, enhancing model robustness to unseen data. However, it's crucial to strike a balance between *Test-Time Mixup* and model performance, as overdoing Mixup may reduce accuracy.

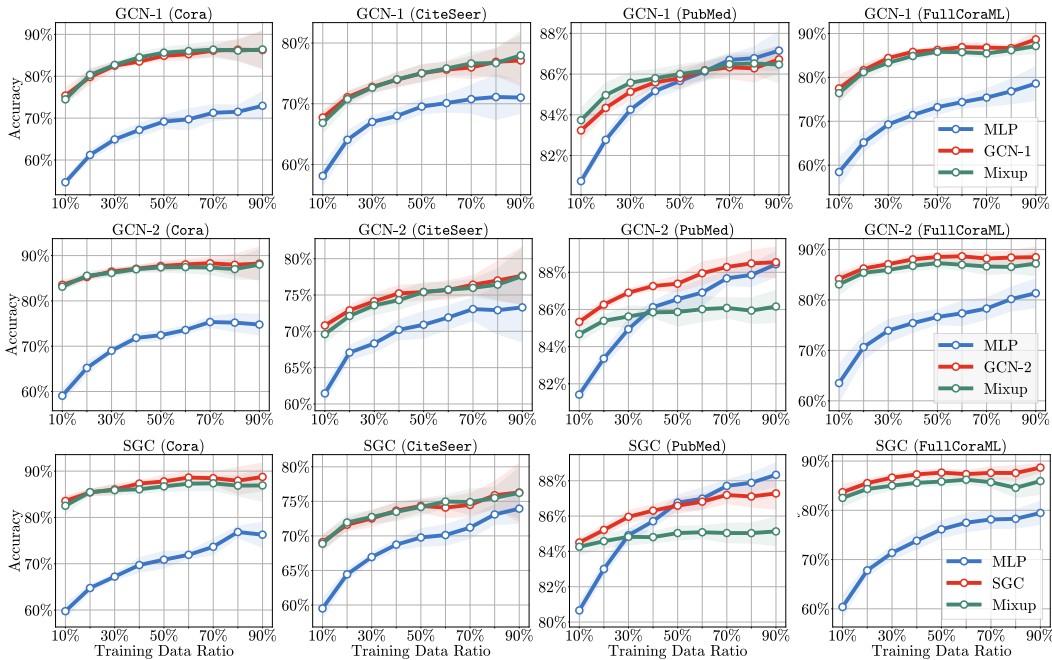

Figure 8: Unifying HMLP and TMLP. these methods together achieve performance comparable to GNNs (despite the PubMed dataset, as the MLPs also exhibit better performance than GNNs.).

## 5 UNIFYING HMLP AND TMLP: VERIFYING THE MAIN RESULTS

In this section, we unify HMLP and TMLP into a single MLP, which integrates both *Homophily Relabel* and *Test-time Mixup*. They are supposed to be equivalent to graph convolution. To unify these two MLPs, we 1) relabel all the nodes in the graph with the mixed label $\hat{Y} = A \cdot Y$. 2) train an MLP on all the nodes with $\hat{Y}$ on relabeled graph, only using node features. 3) employ the well-trained MLP for inference with neighbor aggregation during testing. To test the efficacy of this combined approach, we perform experiments comparing the performance of the unified MLP with that of individual HMLP and TMLP, as well as other GCN. The experimental setup remains consistent with the previous setting with HMLP and TMLP. We present the results in Figure 8. ❺ **Unifying HMLP and TMLP can achieve comparable performance to original GNNs.**

## 6 DISCUSSION AND FUTURE WORK

**Practical Potentials.** Both HMLP and TMLP hold potential value for practical usage. TMLP is training-efficient, as its backbone and training process are both MLP-based; however, the inference requires *Test-Time Mixup* (neighbor aggregation), which can be time-consuming. This limitation also exists in previous works (Han et al., 2022b; Yang et al., 2023). HMLP, on the other hand, is both training and test efficient, as both processes are based on MLP. This suggests the potential of HMLP for practical usage on large-scale graphs, as it eliminates the need for connection information during both training and testing.

**Theoretical Potentials.** HMLP, derived from the Mixup, have the theoretical potential to understand the expressiveness of graph neural network from the Mixup perspective. With the connection between Mixup and graph convolution, HMLP goes beyond traditional methods to understand the learning capability of the graph neural network.

**Future Work.** Based on the above practical and theoretical potentials of the proposed methods in this paper. Here is some potential future works: 1) improve HMLP to make it more practical for large-scale graphs and the performance when the training data ratio is small. 2) The well-studied Mixup strategy would be helpful for understanding the expressiveness of graph neural networks.

## REPRODUCIBILITY STATEMENT

To ensure the reproducibility of our experiments and benefit the research community, we provide the sample source code in the anonymous link `https://anonymous.4open.science/r/GraphConv_is_Mixup-F470`. We will also open-source our implementation after we clean up the codes. The detailed experimental settings required to reproduce our experiments are described in Appendix D.

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

# Appendix

## Table of Contents

## A    RELATED WORKS

In this appendix, we discuss more related work to our paper.

### A.1    GRAPH NEURAL NETWORKS

Starting from graph convolutional network (Kipf & Welling, 2016a), graph neural networks have been applied to various graph learning tasks and show great power for graph learning tasks. Despite the practical power of graph neural network, understanding the generalization and working mechanism of the graph neural networks are still in its infancy (Garg et al., 2020; Zhang et al., 2023a; Yang et al., 2023; Baranwal et al., 2023). The previous works try to understand graph neural networks from different perspectives, such as signal processing (Nt & Maehara, 2019; Bo et al., 2021; Bianchi et al., 2021), gradient flow (Di Giovanni et al., 2022), dynamic programming(Dudzik & Veličković, 2022), neural tangent kernels (Yang et al., 2023; Du et al., 2019; Sabanayagam et al., 2022) and influence function (Chen et al., 2023b). There is also a line of works try to understand graph neural networks by analyzing the connection between GNNs and MLPs (Baranwal et al., 2023; Han et al., 2022b; Yang et al., 2023; Tian et al., 2023).

In our work, we understand graph neural networks through a fresh perspective, Mixup. We believe this work will inspire further research and lead to the development of new techniques for improving the performance and interpretability of GNNs.

### A.2    MIXUP AND ITS VARIANTS

Mixup (Zhang et al., 2018) and its variants (Verma et al., 2019; Yun et al., 2019; Kim et al., 2020; 2021) are important data augmentation methods that are effective in improving the generalization performance of deep neural networks. Mixup is used to understand or improve many machine learning techniques. More specifically, deep neural networks trained with Mixup achieve better generalization (Chun et al., 2020; Zhang et al., 2020; Chidambaram et al., 2022), calibration (Thulasidasan et al., 2019; Zhang et al., 2022a), and adversarial robustness (Pang et al., 2019; Archambault et al., 2019; Zhang et al., 2020; Lamb et al., 2019). As a studies data augmentation method, Mixup can be understood as a regularization term to improve the effectiveness and robustness of neural network (Pang et al., 2019; Archambault et al., 2019; Zhang et al., 2020; Lamb et al., 2019). Mixup are also used to explain some other techniques, such as ensmeble (Lopez-Paz et al., 2023). Additionally, Mixup technique is also used to augment graph data for better performance of GNNs Han et al. (2022a); Wang et al. (2021).

In this work, we use Mixup to explain and understand the graph neural network, and based on the Mixup, we propose two kinds of MLP that achieve almost equivalent performance to graph neural networks. Also, our work is different form the works that chance the GNN training with Mixup, while we connect the graph convolution and Mixup.

### A.3    EFFICIENT GRAPH NEURAL NETWORKS

Although the graph neural networks are powerful for graph learning tasks, they suffer slow computational problems since they involve either sparse matrix multiplication or neighbors fetching during the inference, resulting in active research on the efficiency of graph neural networks. The recent advances in efficient graph neural networks are as follows. Efficient graph neural network (Wang et al., 2023a; Shi et al., 2023; Hui et al., 2023; Zhang et al., 2023b; Zhu et al., 2023b) includes the sampling-based method (Hamilton et al., 2017; Chiang et al., 2019; Zeng et al., 2019), quantization (Liu et al., 2022), knowledge distillation (Zhang et al., 2021; Tian et al., 2023), and graph sparsification (Cai et al., 2020). There is a line of work to try to use MLP to accelerate the GNN training and inference (Zhang et al., 2022b; Wu et al., 2019; Frasca et al., 2020; Sun et al., 2021; Huang et al., 2020; Hu et al., 2021). Simplifying graph convolutional (SGC) (Wu et al., 2019) "simplifying" graph convolutional network by decoupling the neighbor aggregation and feature transformation, the following work such as You et al. (2020) employ the similar basic idea. Zhang et al. (2021) leverage knowledge distillation to transfer the knowledge from GNN to MLP. Han et al. (2022b) and Yang et al. (2023) transfer the weight from MLP to GNN. These kinds of methods use MLP for efficient graph neural networks.

Our proposed methods HMLP and TMLP are efficient for training and/or inference, which can be categorized as an efficient graph neural network. Besides, our proposed method provides a new perspective to understand graph neural networks.

# B    CONNECTION BETWEEN MIXUP AND MORE GRAPH CONVOLUTIONS

## B.1    DISCUSSION ON MORE GRAPH CONVOLUTIONS

We examine the connection between complex graph convolutions used in various graph neural networks and Mixup. We aim to approximately rewrite complex graph convolutions (like those in graph attention networks) into Mixup forms. We present a general Mixup form of graph convolution for node $n_i$ in the graph as follows

$$(\tilde{\mathbf{x}}, \tilde{\mathbf{y}}) = \left( \sum_{j \in \mathcal{N}_i} \lambda_j \mathbf{H}_j, \sum_{j \in \mathcal{N}_i} \lambda_j \mathbf{y}_i \right). \tag{9}$$

Table 3: Examples of the relation between graph convolutions to Mixup at k-layer.

|  | $(\mathcal{N}_i)$ | $(\lambda_j)$ | $(\mathbf{H}_j)$ |
|---|---|---|---|
| GCN | $\mathcal{N}_i$ | $\mathbf{a}_i$ | $\mathbf{H}^k$ |
| SGC | $\mathcal{N}_i^k$ | $\mathbf{a}_i^k$ | $\mathbf{X}$ |
| GAT | $\mathcal{N}_i$ | Attention Score | $\mathbf{H}^k$ |
| PPNP | $\mathcal{N}_i$ | $\mathbf{a}_i$ | Logit |

Equation (9) shows that graph neural networks architectures vary depending on: 1) how to select neighbors set $(\mathcal{N}_i)$, 2) how to select aggregation coefficient $(\lambda_j)$, and 3) what is mixed up $(\mathbf{H}_j)$. For example, if we set $\mathcal{N} = \mathcal{N}_i^2, \lambda = \frac{1}{|\mathcal{N}_i^2|}$, the above equation will be SGC. However, there are a lot of advanced graph neural networks, such as GAT (Veličković et al., 2018)) (attention-based GNN), and PPNP (Gasteiger et al., 2018) (MLP as the backbone). Table 3 shows the relationship between those graph convolutions to Mixup. To include residual connections in some graph convolutions (Chen et al., 2020; Xu et al., 2018), we can expand the neighbors set $\mathcal{N}_i$ of node $n_i$ to include its node representation of the previous time step.

## B.2    EXPERIMENTS ON APPNP

**APPNP** essentially performs the neighbor aggregation in the last layer. Thus, TMLP and HMLP, according to APPNP, are equivalent to GCN. We report the performance in the table below. The results show that TMLP and HMLP achieve better performance than APPNP.

Table 4: Performance Metrics of Different Methods

| MLP | PPNP | TMLP | HMLP |
|---|---|---|---|
| 73.57 | 83.80 | 88.26 | 86.42 |

## B.3    EXPERIMENTS ON APPNP

**GNN with trainable mixing parameter $\lambda$**: In this experiment, we adopted a method similar to GAT to examine the performance of our proposed method, with the mixing parameter $\lambda$ being learned during the training process. The Softmax (Li et al., 2020) is a learnable aggregation operator that normalizes the features of neighbors based on a learnable temperature term. We report the performance in the table below. The results show that the TMLP performed worse than the MLP. This indicates that the learnable mixing parameter $\lambda$ does not work well for our proposed method

Table 5: Performance Metrics of Different Methods

| MLP | GAT-alike | TMLP |
|---|---|---|
| 73.57 | 88.42 | 65.62 |

# C ADDITIONAL EXPERIMENTS

This appendix presents more experiment results on HMLP and TMLP, including 1) comparison experiments of MLP, GCN, and HMLP/TMLP; 2) more experimental results on decision boundary.

## C.1 MORE EXPERIMENTS FOR HMLP

We present further experiments comparing MLP, GCN, and HMLP. These experiments complement those discussed in Figure 4. We expand on that work by evaluating these methods with more GNN architectures. The results are presented in Figure 9.

The results on more GNN architectures indicate that our proposed HMLP method not only outperforms traditional MLP, but also attains a performance that is largely on par with GNNs in the majority of instances.

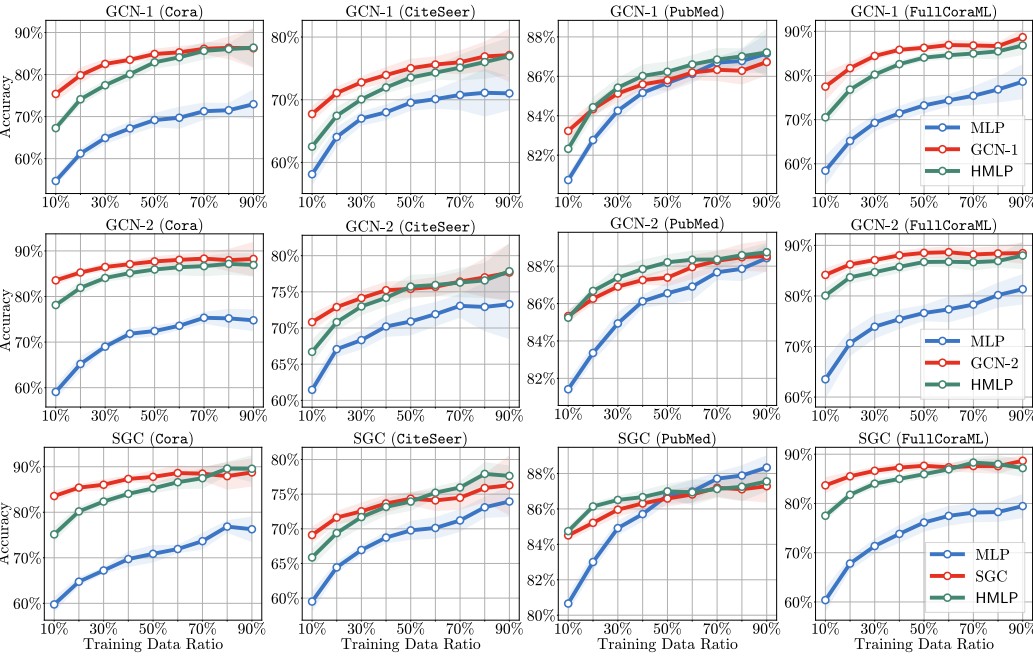

Figure 9: The performance comparison of the GCN, MLP and HMLP (Ours) with GNN architectures. The x-axis represents the ratio of training data, and the y-axis represents the classification accuracy. The results show that our proposed method (HMLP) achieves comparable performance to GCN.

## C.2 MORE EXPERIMENTS FOR TMLP

We present further experiments comparing MLP, GCN, and TMLP. These experiments complement those discussed in Figure 6. We expand on that work by evaluating these methods with more GNN architectures. The results are presented in Figure 10.

With the evaluation of an expanded selection of GNN architectures, the results reveal that our proposed method TMLP typically achieves performance comparable to traditional GNNs in most situations.

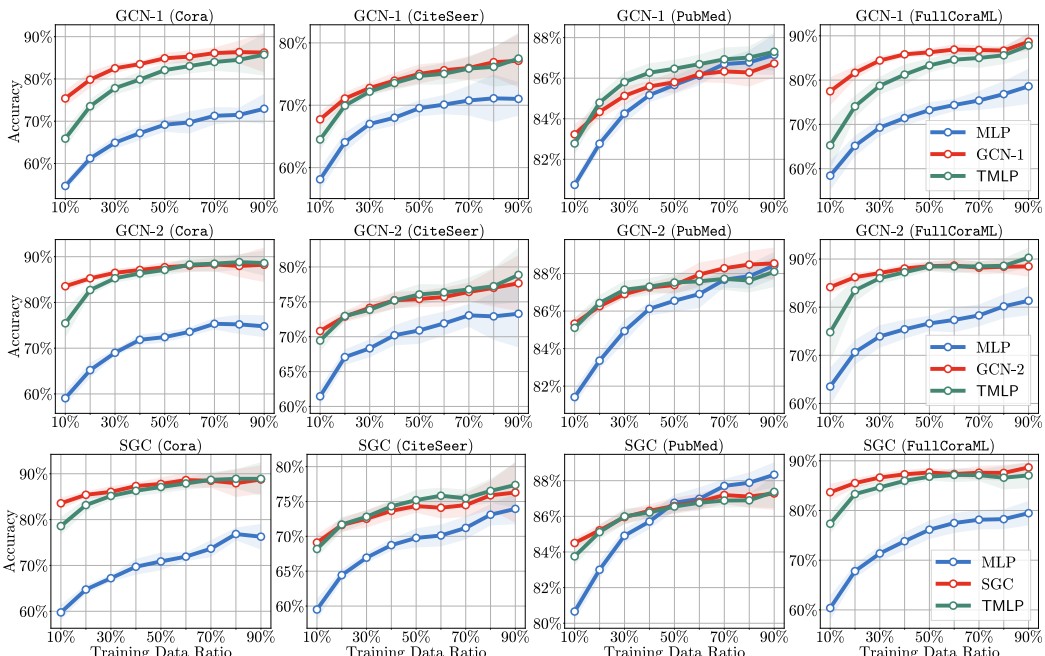

Figure 10: The performance comparison of the GCNs, MLP and TMLP (Ours). GCN-1 is a one-layer GCN, and GCN-2 is a two-layer GCN. The results show that TMLP achieves a comparable performance to GCN and SGC. Note that the architecture is essentially MLP in training time and GCN in the test time.

Table 6: Performance comparison of GNNs and our proposed method with different layers.

|            | 1-layer | 2-layer | 3-layer | 4-layer |
|------------|---------|---------|---------|---------|
| MLP        | 69.11   | 68.95   | 68.39   | 69.22   |
| SGC        | 82.38   | 84.50   | 84.23   | 83.95   |
| HMLP       | 78.56   | 80.41   | 80.96   | 81.15   |
| TMLP       | 82.10   | 83.76   | 83.49   | 83.67   |
| HMLP +TMLP | 83.12   | 83.95   | 84.87   | 84.41   |

## C.3   MORE EXPERIMENTS ON DECISION BOUNDARY

We provide additional experimental results pertaining to the decision boundary of TMLP. Our findings indicate that TMLP tends to shift the node features in the test set away from the pre-learned decision boundary by TMLP.

## C.4   MORE EXPERIMENTS ON DEEPER GCNs

We conducted additional experiments on multi-layer graph neural network (SGC) on the Cora dataset. We use the 20%\40%\40% for train\val\test data split. The test accuracy was determined based on the results from the validation dataset. The test accuracy is reported in the following table. We observe from the results that our proposed HMLP+TMLP method achieves performance comparable to that of SGC when multiple graph convolutions are utilized.

## C.5   EXPERIMENTS ON MORE DATASETS

We conducted additional experiments on more datasets, including FullCora, FullSiteSeer, FullDBLP, and the results are presented in Figure 12. The additional results provide more evidence that graph convolution is a mixup under our proposed conditions.

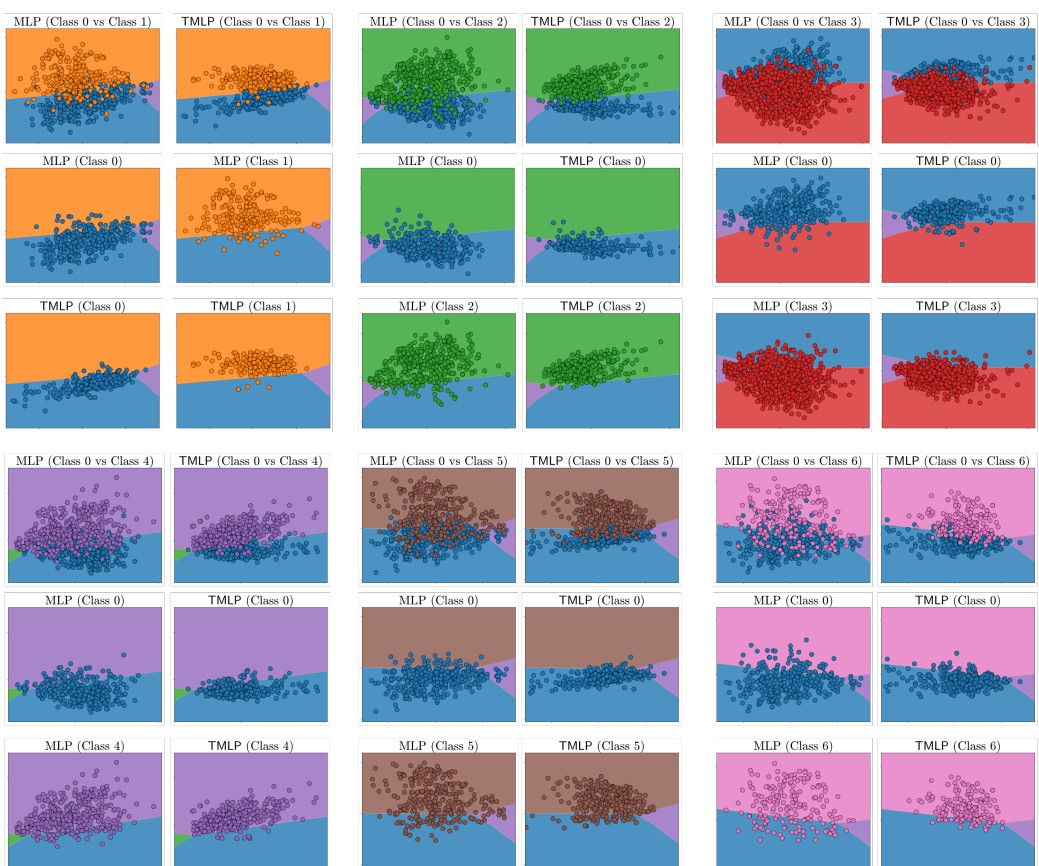

Figure 11: Decision Boundary of MLP and TMLP. TMLP can cause node features to move away from the learned boundary.

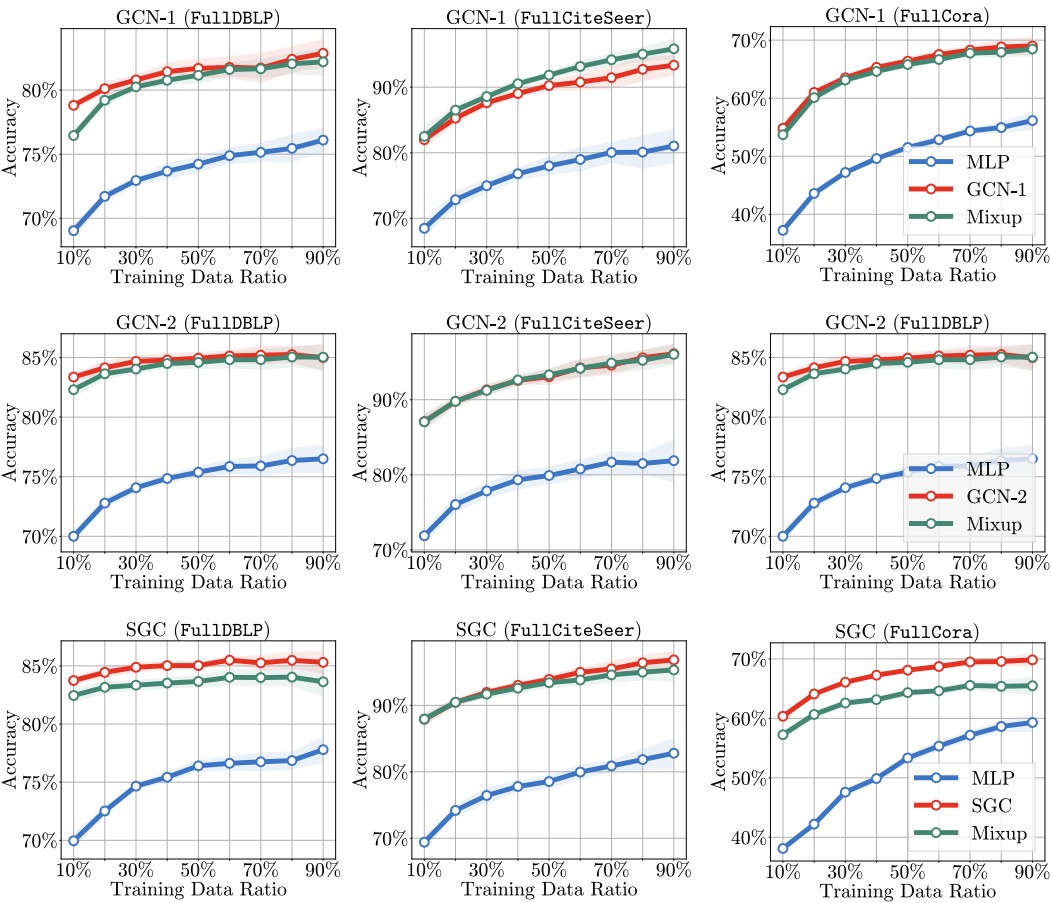

Figure 12: The additional experimental result on unifying HMLP and TMLP. These methods together achieve performance comparable to GNNs on these additional datasets. The additional results show that graph convolution is a mixup under our proposed conditions.

## D EXPERIMENT SETTING

In this appendix, we present the running environment for our experiment and the experimental setup for all the experiments.

### D.1 RUNNING ENVIRONMENTS

We run all the experiments on NVIDIA RTX A100 on AWS. The code is based on PyTorch [2] and torch_geometric [3]. The datasets used in this paper are built-in datasets in torch_geometric, which will be automatically downloaded via the torch_geometric API.

### D.2 EXPERIMENT SETUP FOR FIGURE 4

We provide details of the experimental setup associated with Figure 4. The primary objective of this experiment was to compare the performance of our proposed HMLP model with that of the GCN model. We utilized diverse datasets to facilitate a comprehensive evaluation. We set the learning rate to $0.1$ for all methods and datasets, and each was trained for $400$ epochs.Moreover, we did not use weight decay for model training. The test accuracy reported is based on the best results from the validation set. The node classification accuracy served as the evaluation metric for this experiment.

### D.3 EXPERIMENT SETUP FOR FIGURE 6

We provide details of the experimental setup associated with Figure 6. We set the learning rate to $0.1$ for all methods, and each was trained for $400$ epochs. Additionally, weight decay was not utilized in this process. The test accuracy reported is based on the best results from the validation set. The node classification accuracy served as the evaluation metric for this experiment.

### D.4 EXPERIMENT SETUP FOR FIGURE 8

We provide details of the experimental setup associated with Figure 8. We set the learning rate to $0.1$ for all methods, and each was trained for $400$ epochs. Additionally, weight decay was not utilized in this process. The test accuracy reported is based on the best results from the validation set. The node classification accuracy served as the evaluation metric for this experiment.

## E DATASETS

In this experiment, we use widely adopted node classification benchmarks involving different types of networks: three citation networks (Cora, CiteSeer and PubMed). The datasets used in this experiment are widely adopted node classification benchmarks. These networks are often used as standard evaluation benchmarks in the field of graph learning.

- Cora: is a widely used benchmark data for graph learning, which is a citation network, where each node represents a research paper, and edges denote the citation relation. Node features are extracted from the abstract of the paper.

- CiteSeer is another citation network used frequently in GNN literature. It follows a similar structure to the Cora dataset, where each node represents a scientific document, and the edges are citation information between them. The node features are derived from the text of the documents.

- PubMed: PubMed is a citation network composed of scientific papers related to biomedicine. As with the other datasets, nodes represent documents, and edges indicate citation relationships. Node features correspond to term frequency-inverse document frequency (TF-IDF) representations of the documents.

---

[2] https://pytorch.org/
[3] https://pyg.org/

- `FullCoraML` is a subset of the full `Cora` dataset. It retains the same structure as the `Cora` dataset but is restricted to documents and citations within the field of machine learning. The node features are the bag-of-words representation for the papers.

## F    BROADER IMPACT

This work uncovers the connection between graph convolution and Mixup, which is anticipated to have significant implications in the graph learning field. The potential impacts include: 1) offering a fresh perspective for a deeper understanding of graph convolution and 2) providing the potential to foster the development of more efficient graph neural networks. Our work contributes both practical and theoretical advances to the AI community, particularly within the domain of graph neural networks (GNNs).

- **Efficient Alternative Training and Inference Method for GNNs on Large Graph Data.** HMLP presents an efficient solution for both training and testing on large-scale graphs, as it completely eliminates the need for connection information during both stages. TMLP, though also MLP-based, emphasizes training efficiency. The improved training efficiency of TMLP can still be advantageous for reducing the training cost on large graph.

- **Opening a New Door for Understanding Graph Convolution.** The introduction of mixup, a simple node-level operation, offers a novel perspective on graph convolution by representing it in a mixup form. Traditional graph convolution employs the adjacency matrix of the entire graph, a concept that can be complex to comprehend or articulate. By approaching the understanding of graph neural networks from the entire graph level, we can demystify these complexities. This presents an essential avenue for exploration in future work related to our paper.

