# OpenReview forum: "On the Equivalence of Graph Convolution and Mixup"
_ICLR.cc/2024/Conference — ICLR 2024 Conference Withdrawn Submission_

### Official Review · Reviewer_ocCu · 2023-10-30

**Soundness:** 2 fair
**Presentation:** 3 good
**Contribution:** 1 poor
**Rating:** 3
**Confidence:** 3

**Summary:**

The paper studies the relationship between Graph neural networks and Mixup and formally shows that graph convolution is equivalent to Mixup under certain conditions. The paper proposed two variants of MLPs based on these findings that match the performance of GNNs on node labeling tasks.

**Strengths:**

The paper is investigating an interesting relationship between graph convolution and Mixup and has several strengths:

1. The paper is very detailed and convincing about the similarities of GNNs and Mixup.
2. The presentation quality is good.
3. The proposed methods are potentially valuable (for example, can be fast to train and evaluate for large graphs).

**Weaknesses:**

While the paper is convincing about the equivalence of graph convolution and Mixup, it's contributions are quite limited.

1. The paper claims potential speed ups (e.g. "this work ... facilitating efficient training and inference for GNNs when dealing with large-scale graph data"), but does not provide any time/memory cost comparisons for training/inference. Therefore, it's hard to be convinced about the benefit of the proposed approaches. For example, see Table 1 in [Huang2021] and results in [Duan2022] for such a comparison.

2. Experiments are performed only on Cora, CiteSeer, and PubMed, which are relatively small-scale and sensitive to splits and other factors. Moreover, on PubMed a simple MLP is very strong questioning the usefulness of this dataset for GNNs. It would be more useful to evaluate on more large-scale datasets from Open Graph Benchmark and/or other datasets used in related papers like [Huang2021, Duan2022].

3. The main baselines in this paper are MLP, GNN and SGC. However, a more interesting comparison would be to [Huang2021] that this paper cites but does not discuss in detail and/or other methods like those compared in [Duan2022].

**References:**

- [Huang2021] Huang et al. "Combining label propagation and simple models out-performs graph neural networks", ICLR 2021.
- [Duan2022] Duan et al. "A Comprehensive Study on Large-Scale Graph Training: Benchmarking and Rethinking", NeurIPS Track on Datasets and Benchmark 2022.

**Questions:**

Some previous papers like [Wang2020] also studied the connection between Label Propagation and GNNs. Do the authors think Mixup is connected to Label Propagation and some insights can be derived from that?

[Wang2020] Wang et al. Unifying Graph Convolutional Neural Networks and Label Propagation, 2020

---

### Official Review · Reviewer_12Mm · 2023-11-02

**Soundness:** 3 good
**Presentation:** 3 good
**Contribution:** 3 good
**Rating:** 5
**Confidence:** 3

**Summary:**

This interesting paper establishes the theoretical link between MixUp and Graph-Convolutions.
This has the potential to speed-up graph convolutions by transforming them in simple DNN's + MixUp (+H-relabeling).

**Strengths:**

By examining the equations of both MixUp and GCN authors derive the link between the two.
In particular a relabeling of the neighboring nodes allow to express GCN's as DNN's + MixUp.
I believe this is a pretty interesting conclusion!
As mentioned this has the potential to speed up computation / scale of GCN's by dropping (expensive) sparse primitives and replacing them with efficient dense operations.
The paper is very extensive and well written with lots of good plots and examples!

**Weaknesses:**

I may have missed something but I believe there is a loss when going from GCN's to DNN+MixUp - loss in structural generalization?
(see questions)

**Questions:**

While I understand that DNN+MixUp(+H-relabel) can contribute to same/similar computational graph as DNN's - as showed theoretically and practically by the authors - I also believe that GCN strength is that they can generalize to different graph topologies.

The neighbour reduction operator of GCN is shape independent - and as such it allows to change graph structure among samples/batches as well as generalize (to some extent) between train and test sets.

I struggle to see how the proposed method can achieve that. But I may have missed something - please explain.

In which case it should be explained in the text (the lack of the structural generalization feature - constraining to use same graph topology).

---

### Official Review · Reviewer_Psnu · 2023-11-03

**Soundness:** 2 fair
**Presentation:** 3 good
**Contribution:** 1 poor
**Rating:** 3
**Confidence:** 4

**Summary:**

The paper studies the connection between graph convolution and Mixup: the neighboring feature-aggregation mechanism of graph convolutional networks highly resembles the feature/label-averaging of Mixup. Given this observation, the paper shows that graph convolution is equivalent to Mixup under two conditions, which are 1) Homophily Relabel - relabeling each node based on its neighbors' nodes prior to training and 2) Test-Time Mixup - aggregating neighboring features during test time. The equivalence is verified empirically by training an MLP (that is oblivious of the graph structure) under Homophily Relabel (HMLP) or Test-Time Mixup (TMLP) and showing that their performance is comparable to that of GCNs. Further investigating the node representation space and the decision boundaries within also reveal that TMLP leads to more discriminative representations with better robustness to unseen data.

**Strengths:**

- **Novel and original.** The paper proposes a very interesting interpretation of graph convolution, which to the best of my knowledge, has not been studied yet.
- **Great presentation.** Paper is well-written with clear mathematical derivations that are easy to follow.

**Weaknesses:**

- **Overall significance is unclear and claimed practical/theoretical implications lack support.**
  - [W1] While Section 6 mentions how TMLP and HMLP can be training-efficient with large-scale graphs since they are MLP-based, there are no experiments to compare computational costs and demonstrate this claim. I also suspect the gain in efficiency would be fairly small since graph convolution can be highly optimized via sparse-dense matrix multiplications.
  - [W2] The theoretical implications also seem a bit far-fetched: there is a rich line of work on the theoretical expressivity of GNNs [A, B, C, D, E], yet the paper does not convey any clear connection between Mixup and such work to support its theoretical significance. This could most likely constitute another research paper on its own, but any further discussion on what makes "the Mixup perspective" particularly interesting with respect to GNN expressivity would be helpful.
- **Discussion on empirical results do not provide much insight and are often unconvincing due to lack of evidence.**
  - [W3] Section 3.2 and Figure 3 show that HMLP and GCN share the same test curves, but not the training curves, which the paper conjectures to be "because the labels of nodes are changed". I think the discussion could go a bit deeper (maybe via proper ablation),  considering that the training dynamics of HMLP differs from that of GCN in ways other than the node-labels: not only are all the node-labels mixed, but also 1) nodes in the test-set are used for training HMLP (not the case for GCN) and 2) the model parametrization is different.
  - [W4] Section 4.2 mentions how "overdoing Mixup may reduce accuracy", but this observation is not evident from the results in Figure 7, nor is the discussion consistent with the claim that "Test-Time Mixup enhances model robustness". Are there any experimental results that can support this observation? Is this discussion in any way related to the oversmoothing problem of graph convolution [F]?
  - [W5] Figure 8 shows a fairly large discrepancy between GNNs and Mixup on the PubMed dataset. Considering that the main claim is that GNNs and Mixup are equivalent, why are we seeing this discrepancy on performance despite PubMed being a homophilic network just like Cora and CiteSeer?

[A] Morris et al., Weisfeiler and Leman Go Neural: Higher-order Graph Neural Networks. (AAAI 2019)\
[B] Xu et al., How Powerful are Graph Neural Networks? (ICLR 2019)\
[C] Maron et al., Provably Powerful Graph Networks. (NeurIPS 2019)\
[D] Zopf, 1-WL Expressiveness Is (Almost) All You Need. (IJCNN 2022)\
[E] Feng et al., How Powerful are K-hop Message Passing Graph Neural Networks. (NeurIPS 2022)\
[F] Li et al., Deeper Insights into Graph Convolutional Networks for Semi-Supervised Learning. (AAAI 2018)

**Questions:**

- [Q1] I am not sure whether Section 2.4 is really necessary on its own as it does not seem to provide significant information. The section could instead be merged to the Related Work paragraph in Section 1.
- [Q2] Question on HMLP: given the example graph in Figure 2, HMLP uses $(x_i, (0.5, 0.5))$ for training, yet the ground-truth sample during test time will be just one of $(x_i, (1, 0))$ or $(x_i, (0, 1))$. In this case, wouldn't the difference in the labels be detrimental for test-time performance since the model will be trained to output $(0.5, 0.5)$ given $x_i$?
- [Q3] How are the "MLP"s and "Mixup"s different for each row of plots in Figure 8? Was slightly confused why the same method appears in different graphs with different accuracies.
- [Q4] The end of Practical Potentials paragraph in section 6 mentions that HMLP "eliminates the need for connection information during both training and testing". Is this true? Although the model being trained may not require connectivity information, the Homophily Relabeling taken place prior to training does seem to require the connectivity.

There are also several typos:
- End of Section 1: "be considered as a generlized ..." -> "be considered as a generalized ..."
- Notations paragraph in Section 2: "one-hot labels $Y \in \\{0,1\\}^{N \times (C-1)}$..." -> Shouldn't it be $Y \in \\{0,1\\}^{N \times C}$ with $C$ classes?
- Top of page 5: "We rewrite the $\tilde{a}i \cdot X$..." -> should be subscript "$\tilde{a}_i \cdot X$"
- Section 2.4: "... understanding of this phoneme ..." -> "... understanding of this phenomenon ..."
- End of Section 3.1: "Thus in the next, we propose ..." -> "Thus in the next paragraph, we propose ..."?
- Bottom of page 7: "In the Figure 4, the performance ..." -> Wrong reference. "In Figure 6, the performance ..."
- End of Section 4.1: "The node representation clearly show that what TMLP ..." -> "... clearly show what TMLP ..."